# Therapeutic Options for Recurrent Glioblastoma—Efficacy of Talaporfin Sodium Mediated Photodynamic Therapy

**DOI:** 10.3390/pharmaceutics14020353

**Published:** 2022-02-02

**Authors:** Tatsuya Kobayashi, Masayuki Nitta, Kazuhide Shimizu, Taiichi Saito, Shunsuke Tsuzuki, Atsushi Fukui, Shunichi Koriyama, Atsushi Kuwano, Takashi Komori, Kenta Masui, Taketoshi Maehara, Takakazu Kawamata, Yoshihiro Muragaki

**Affiliations:** 1Department of Neurosurgery, Tokyo Women’s Medical University, 8-1 Kawadacho, Shinjuku-ku, Tokyo 162-8666, Japan; opera58840428@gmail.com (T.K.); taiichis@gmail.com (T.S.); st09212005@yahoo.co.jp (S.T.); fukui.atsushi@twmu.ac.jp (A.F.); song.4.u.01.16@gmail.com (S.K.); cho_hunseki@yahoo.co.jp (A.K.); kawamata.takakazu@twmu.ac.jp (T.K.); ymuragaki@twmu.ac.jp (Y.M.); 2Faculty of Advanced Techno-Surgery, Tokyo Women’s Medical University, 8-1 Kawadacho, Shinjuku-ku, Tokyo 162-8666, Japan; 3Department of Neurosurgery, Massachusetts General Hospital, Harvard Medical School, 185 Cambridge Street, Boston, MA 02114, USA; KSHIMIZU3@mgh.harvard.edu; 4Department of Neurosurgery, Tokyo Medical and Dental University, 1-5-45 Yushima, Bunkyo-ku, Tokyo 113-8519, Japan; maehara.nsrg@tmd.ac.jp; 5Department of Laboratory Medicine and Pathology (Neuropathology), Tokyo Metropolitan Neurological Hospital, 2-6-1 Musashidai, Fuchu-shi, Tokyo 183-0042, Japan; komori-tk@igakuken.or.jp; 6Department of Pathology, Tokyo Women’s Medical University, 8-1 Kawadacho, Shinjuku-ku, Tokyo 162-8666, Japan; masui-kn@twmu.ac.jp

**Keywords:** recurrent glioblastoma, photodynamic therapy, talaporfin sodium, photosensitizer, lower grade glioma

## Abstract

Recurrent glioblastoma (GBM) remains one of the most challenging clinical issues, with no standard treatment and effective treatment options. To evaluate the efficacy of talaporfin sodium (TS) mediated photodynamic therapy (PDT) as a new treatment for this condition, we retrospectively analyzed 70 patients who underwent surgery with PDT (PDT group) for recurrent GBM and 38 patients who underwent surgery alone (control group). The median progression-free survival (PFS) in the PDT and control groups after second surgery was 5.7 and 2.2 months, respectively (*p* = 0.0043). The median overall survival (OS) after the second surgery was 16.0 and 12.8 months, respectively (*p* = 0.031). Both univariate and multivariate analyses indicated that surgery with PDT and a preoperative Karnofsky Performance Scale were significant independent prognostic factors for PFS and OS. In the PDT group, there was no significant difference regarding PFS and OS between patients whose previous pathology before recurrence was already GBM and those who had malignant transformation to GBM from lower grade glioma. There was also no significant difference in TS accumulation in the tumor between these two groups. According to these results, additional PDT treatment for recurrent GBM could have potential survival benefits and its efficacy is independent of the pre-recurrence pathology.

## 1. Introduction

Glioblastoma (GBM) is one of the most malignant primary brain tumors with a quite poor prognosis. The standard treatment for newly diagnosed GBM is maximal surgical resection followed by radiotherapy with concomitant and adjuvant temozolomide-based chemotherapy. Nevertheless, the median survival time for patients who complete the standard treatment is only about 15 months or less [1]. One factor that contributes to the unsatisfactory prognosis in GBM is that most recur following standard treatment. On recurrence, the therapeutic options include surgical rechallenge, additional existing chemotherapy, and additional radiation, but their efficacy is limited. The median overall survival (OS) after various retreatments for recurrent GBM has been reported at 6.5–7.6 months [2,3,4]. There is currently no standard treatment for recurrent GBM, because no existing therapy has demonstrated superiority [3,5,6]. Therefore, new effective treatments for this condition are urgently required.

Photodynamic therapy (PDT) uses specific wavelengths of light to activate photosensitizers accumulated in the tumor. When the photosensitizer is activated, it generates reactive oxygen species (ROS) in the local irradiated area. The ROS destroy various cellular organelles and tumor blood vessels, thereby destroying the tumor itself [7]. There are several types of photosensitizers, each with different biological and pharmacokinetic characteristics, including subcellular localization, excitation and emission wavelength, the mechanism for inducing cell death, and the uptake or clearance level of the specific tissue [8,9].

Recently in Japan, talaporfin sodium (mono-L-aspartyl chlorine, NPe6, TS), a chlorin-based photosensitizer has been clinically applied in PDT for lung and esophageal cancers, and in malignant brain tumors [10,11,12]. In a phase II trial of talaporfin sodium mediated PDT (TS-PDT) for malignant brain tumors, the median progression-free survival (PFS) and the median OS for newly diagnosed GBM were 12.0 and 24.8 months, respectively [12]. Furthermore, we recently performed a retrospective analysis of the efficacy of intraoperative TS-PDT for newly diagnosed GBM and reported that the TS-PDT group had a significantly better PFS of 19.6 months and OS of 27.4 months compared with standard treatment [13]. Consideration of these favorable results led TS-PDT to be substantiated as a valuable additional treatment for newly diagnosed GBM. On the other hand, there are no cohort or case-control studies of TS-PDT for recurrent GBM, and its efficacy is still unclear.

Given these current circumstances, as the second report on the clinical efficacy of TS-PDT for a malignant brain tumor, we analyzed the prognostic data of single-center experience cases and evaluated the therapeutic effect on recurrent GBM in this study. To analyze tumor pathology and pharmacokinetics of TS, we also examined the amount of TS uptake in recurrent glioblastoma and verified whether the effect of TS-PDT depended on previous pathological results before recurrence.

## 2. Materials and Methods

### 2.1. Patient Selection and Treatment Criteria

In this single-center retrospective analysis, 70 consecutive patients who underwent surgical resection and intraoperative TS-PDT (PDT group) for recurrent malignant glioma between February 2014 and December 2018 were compared with 38 consecutive patients with recurrent GBM who underwent surgical resection alone (control group) during the same period. At our institution, patients with a Karnofsky Performance Status (KPS) score ≥ 60 were considered for reoperation if they developed recurrence of malignant glioma after standard multidisciplinary treatment. The indication for surgery was restricted to cases in which total resection of the recurrent contrast-enhancing lesion was considered feasible. For patients with a KPS score of 40 or 50 and a strong desire for surgery, the surgical indication was limited to those expected to experience an improvement in clinical symptoms with reoperation in addition to the previously stated conditions. This study was approved by the Institutional Review Board of our institution (approval code: 3540-R6). 

### 2.2. Intraoperative TS-PDT Protocol

Intraoperative TS-PDT for recurrent GBM was performed using the same protocol that we previously reported for newly diagnosed GBM and other malignant brain tumors in Japan [12,13,14]. Patients received a single intravenous injection of TS at a dose of 40 mg/m^2^, 22–26 h before surgery. After maximal resection of the contrast-enhanced lesions, laser irradiation to the resection cavity was performed using a 664 nm semiconductor laser (PD laser BT, Meiji Seika Co., Ltd., Tokyo, Japan) with an irradiation power density of 150 mW/cm^2^ and an irradiation energy density of 27 J/cm^2^ within a circle (diameter: 1.5 cm) per location. The irradiation was performed to cover the entire resection cavity without overlapping of the irradiation fields.

### 2.3. Neuropathological Analysis

Histopathological diagnosis was conducted based on the WHO guidelines of 2007 and 2016 [15,16]. For patients diagnosed according to the 2007 WHO classification, IDH mutation was retrospectively analyzed and re-diagnosed using the 2016 WHO classification. IDH mutation status was examined by immunohistochemistry using R132H-specific antibody (DIA-H09, Dianova GmbH, Hamburg, Germany). In case of negative results, direct DNA sequencing of the tumor sample was additionally performed. Mib-1 index was assessed using immunohistochemistry with Mib-1 monoclonal antibody (M7240, Agilent Technologies, Santa Clara, CA, USA). The presence of 1p/19q codeletion was analyzed using fluorescence in situ hybridization. The methylation status of O-6-methylguanine-DNA methyltransferase (MGMT) was not evaluated, but alternatively, the expression of MGMT protein was determined by conducting immunohistochemistry with anti-MGMT monoclonal antibody (MAB16200, Merck, Darmstadt, Germany).

### 2.4. Evaluation of TS Uptake in Recurrent GBM

TS exhibits a soret absorption band at ~400 nm and produces emission at a wavelength light of ~660 nm after excitation [17]. A fluorescence microscope (BZ-X710, KEYENCE, Osaka, Japan) at an excitation wavelength of 400 nm to image and evaluate the uptake of TS in recurrent GBM samples. To quantify the TS uptake in recurrent GBM, we measured the peak fluorescence intensity from each tumor sample using a semiconductor laser unit (LDS1005BL, Precise Gauges Co., Ltd., Shizuoka, Japan) according to a previously reported method [18] and compared the relationship between TS uptake and previous pathology before recurrence.

### 2.5. Patient Assessment and Follow-Up

All patients were evaluated with 0.4T intraoperative MRI images (APERTO Lucent, Hitachi, Ltd., Tokyo, Japan) before and after tumor removal or 1.5T MRI images during the early postoperative period (within 72 h after surgery). Based on these MRI images, the extent of resection (EOR) of the contrast-enhanced lesion was categorized as follows: Gross total resection (GTR) was considered for an EOR > 98%, subtotal resection (STR) 95–98%, and partial resection (PR) < 95%. An additional postoperative MRI was performed 2 weeks after surgery, followed by imaging every month; assessment of tumor recurrence was determined based on the Response Assessment in Neuro-oncology (RANO) criteria [19]. The severity of adverse events was determined based on the Common Terminology Criteria for Adverse Events (CTCAE) version 5.0. 

### 2.6. Statistical Analysis

All statistical analyses were performed using EZR version 1.54 (Saitama Medical Center, Jichi Medical University, Saitama, Japan) [20], which is a graphical user interface for R (The R Foundation for Statistical Computing, Vienna, Austria). 

For intergroup comparison, the Mann–Whitney U test was used for continuous variables and the χ^2^ test for categorical variables. Time-to-event analysis was performed using Kaplan–Meier curves and log-rank tests. PFS was defined as the time from the date of operation for recurrent GBM to the date of documented evidence of tumor progression according to the RANO criteria. OS was defined as the time from the date of surgery for recurrent GBM to the date of death or censoring at the last known date alive. Univariate and multivariate analyses were performed using the Cox proportional hazards model. Statistical significance was set at a *p* value < 0.05 and all reported *p*-values are two-sided.

## 3. Results

### 3.1. Patient Demographics and Characteristics

The demographics and clinicopathological characteristics of patients in the PDT and control groups were described in Table 1. Among the 108 patients, the PDT group comprised 70 patients (male 56%; female 44%), with the median age at reoperation being 43.5 (range 20–80) years. The control group contained 38 patients (male 68%; female 32%), with a median age at reoperation of 42 (range 16–71) years. There were no significant differences regarding age and sex between the two groups (*p* = 0.32 and 0.22, respectively), and there was no significant difference in preoperative KPS (*p* = 0.063). In the PDT group, results of the EOR evaluation were GTR 91.4%, STR 5.7%, PR 2.9%, and in the control group they were GTR 94.7%, PR 5.3% (*p* = 0.37). The histopathological results revealed 69 cases of GBM and one case of gliosarcoma in the PDT group, and all 38 cases were GBM in the control group. Of the 69 patients in the PDT group with GBM, 43 had already been previously diagnosed with GBM based on pathology before recurrence, and 26 had been diagnosed as lower grade glioma (LGG). IDH1 R132H mutation was identified in 22.9% of the PDT group and 39.5% of the control group (*p* = 0.11). The median Mib-1 index in the PDT and control groups were 17.0 (range 1.6–51.4) and 20.7 (range 4.0–46.8), respectively, with no significant difference (*p* = 0.11). There was no 1p/19q codeletion in any patient, and there was no difference in O-6-Methylguanine-DNA methyltransferase (MGMT) protein expression between the two groups (*p* = 0.73).

### 3.2. Patient Safety

The complication rate was 4.3% (3 patients) in the PDT group and 0% (0 patients) in the control group (*p* = 0.55). In the PDT group, one patient experienced wound dehiscence (grade 3) and required surgical reconstruction, one patient had cerebrospinal fluid leakage (grade 2), and one patient had acute epidural hematoma as postoperative hemorrhage (grade 3) and required surgical treatment. No other adverse events ≥ grade 3 according to the CTCAE version 5.0, were observed in both groups.

### 3.3. Survival Analysis

The median PFS after surgery for recurrence of the 70 patients in the PDT group was 5.7 months (95% confidence interval [CI] 3.4–7.1), and the median PFS in the control group was 2.2 months (95% CI 1.5–4.0); the PDT group exhibited significantly longer PFS than the control group (*p* = 0.0043, Figure 1A). The median OS after surgery for recurrence in the PDT group was 16.0 (95% CI 13.7–22.5) months, the 1-year OS rate was 73% and the 2-year OS rate was 37.4%, whereas the median OS in the control group was 12.8 (95% CI 9.3–15.0) months, the 1-year OS rate was 58.8%, and the 2-year OS rate was 11.5%; the PDT group exhibited better OS than the control group which was statistically significant (*p* = 0.031, Figure 1B).

In the PDT group, there were 43 patients whose previous pathology before recurrence was GBM (GBM group), and 26 patients whose previous pathology before recurrence was lower-grade glioma (LGG group). The median PFS after recurrence in the GBM and LGG groups were 6.3 (95% CI 3.1–8.4) and 4.2 (95% CI 2.7–6.9) months, respectively (*p* = 0.31, Figure 2A). The median OS after recurrence in the GBM and LGG groups were 15.4 (95% CI 13.4–31.6) and 18.3 (95% CI 11.9–33.4) months, respectively (*p* = 0.91, Figure 2B). Therefore, PFS and OS exhibited no significant between-group differences.

### 3.4. Univariate and Multivariate Analysis

The relationship between PFS, OS, and prognostic factors such as age, preoperative KPS score, pre-recurrence pathology, IDH mutation, and addition of PDT were examined in univariate and multivariate analyses. The results revealed that preoperative KPS score and the addition of PDT were independent and significant prognostic factors in both univariate and multivariate analyses for PFS (Table 2). Similarly, both univariate and multivariate analyses showed that preoperative KPS score and addition of PDT were significant independent prognostic factors for OS (Table 3). In contrast, IDH mutation and pre-recurrence pathology were not significant prognostic factors in this study.

### 3.5. TS Uptake Comparison

TS uptake in recurrent GBM and peri-tumoral normal tissue samples was evaluated and photographed using fluorescence microscopy during the surgery (Figure 3). There is greater TS accumulation in recurrent GBM. Among the samples for which intraoperative uptake could be verified, the degree of uptake in seven cases of recurrent GBM was quantitatively compared by classifying them into GBM and LGG groups based on pre-recurrence pathology (Figure 4). As a result, 14 samples from five cases in the GBM group and six samples from two cases in the LGG group were analyzed, and there was no significant difference in the fluorescence intensity for each group (*p* = 0.20).

### 3.6. Representative Cases


Case 1


A 53-year-old woman presented with sudden onset seizure and was hospitalized. MRI revealed a fluid-attenuated inversion recovery (FLAIR) high lesion in the right frontal lobe with no enhancement on gadolinium (Figure 5A,B). Gross-total removal of the FLAIR high lesion was performed via awake craniotomy (Figure 5C,D), and the pathological diagnosis was IDH wildtype anaplastic astrocytoma. The patient received fractionated radiation therapy (60 Gy) and concomitant temozolomide-based chemotherapy followed by maintenance temozolomide-based chemotherapy for another 5 courses. Nine months after the initial operation, follow-up MRI detected a recurrence of the tumor around the removal cavity (Figure 5E,F). Gross-total removal of the enhanced lesion was performed, and the cavity wall was irradiated in four spots (Figure 5G,H). The pathological diagnosis was GBM. Postoperatively, the patient transiently exhibited mild manual dexterity and dysphagia. The MRI on day 14 following surgery revealed fluid collection and slight edema (Figure 5I,J), and these findings partially resolved within 2 months (Figure 5K,L). Maintenance temozolomide-based chemotherapy was resumed and continued for 24 courses. At the latest follow-up of 47 months after the second surgery, the MRI demonstrated no recurrence and the patient had a KPS score of 90 (Figure 5M,N).


Case 2


A 59-year-old woman presented with aphasia and speech disturbance. MRI revealed a round mass lesion in the left temporal lobe. The tumor exhibited low intensity on T1-weighted images, ring-like enhancement on gadolinium uptake (Figure 6A), and high intensity on FLAIR images (Figure 6B). Gross-total removal of the enhanced lesion was performed (Figure 6C,D), and the pathological diagnosis was IDH wildtype GBM. The patient received fractionated radiation therapy (60 Gy) and concomitant temozolomide-based chemotherapy following autologous formalin-fixed tumor vaccine (AFTV). Three months after the initial operation, follow-up MRI detected tumor recurrence in the anterior part of the removal cavity (Figure 6E,F). Gross-total removal of the enhanced lesion was performed again, and the cavity wall was irradiated in six spots (Figure 6G,H). The pathological diagnosis was GBM. Postoperatively, the patient still exhibited mild aphasia; the symptoms were unchanged compared to before the surgery. The 14-day postoperative MRI revealed fluid collection and edema (Figure 6I,J), and these resolved within 2 months (Figure 6K,L). Five months after the second operation, an enhanced lesion appeared in the medial part of the cavity wall (Figure 6M), and it exhibited uptake in a methionine PET study with a tumor tissue/normal tissue ratio of 2.74 (Figure 6N). Removal of the enhanced lesion was performed, and the cavity wall was irradiated again in six spots. The pathological diagnosis was necrotic tissue, and there was no evidence of tumor recurrence (Figure 6Q,R). At the latest follow-up, 50 months after the second surgery, the MRI showed no recurrence and the patient had a KPS score of 80 (Figure 6O,P).

## 4. Discussion

Although the first uses of light as a therapeutic agent date back many centuries, the evolution of PDT as a cancer treatment occurred ~1940–1950 with the discovery and purification of hematoporphyrin derivatives, which were highly accumulative in cancerous tissue [21]. Since then, a large number of photosensitizers have been developed for PDT against tumors [22]. To date, studies of PDT on tumors have shown that the mechanism of tumor cell death and destruction varies depending on the type of photosensitizer and the irradiation conditions including a combination of apoptosis, necrosis, autophagy, necroptosis, parthanatos or other regulated cell death, immunogenic cell death, and cell death due to microvascular damage or occlusion [23,24,25,26,27]. In addition to brain tumors, several studies on photosensitizers have demonstrated efficacy in various cancers such as lung, esophageal, head and neck, otorhinolaryngological, skin, hepatobiliary, pancreatic, colorectal, prostate, bladder, and ovarian cancers [9,28]. Despite several potential candidates as photosensitizers for use in the treatment of brain tumors, a limited number are currently being used clinically based on the results of clinical trials and side effects such as photosensitivity [29,30]. In recent years, the photosensitizers used in clinical trials for malignant brain tumors are mainly TS [31] and 5-ALA [32,33,34], both of which exhibit a favorable safety profile.

Tumor accumulation rate is known to vary depending on the photosensitizer, and previous in vivo experiments demonstrated that the uptake level of TS in the brain tumor was 23.1 times that of normal brain tissue, which was also 7.78 times higher than the uptake of 5-ALA and 13 times higher than that of Photofrin [35,36]. Therefore, TS-PDT was expected to be a useful additional treatment for malignant brain tumors. In fact, a previously reported comparative retrospective analysis of surgical resection with TS-PDT in 30 consecutive cases of newly diagnosed GBM and 164 consecutive cases treated with surgical resection alone during the same period showed a satisfactory result [13]. Contrarily, while the efficacy of TS-PDT in recurrent malignant brain tumors is an important clinical question, there has only been one clinical report in a limited number of patients [14].

In this study, we focused on recurrent GBM and evaluated the efficacy of TS-PDT. As shown in the time-to-event analysis, the addition of TS-PDT to surgical treatment in recurrent GBM significantly improved the prognosis compared with surgical treatment alone and the efficacy of TS-PDT was independent of the pre-recurrence pathology. The PFS curve of the PDT group decreased relatively early; subsequently, a certain number of patients exhibited long-term PFS. We previously reported that tumors can exhibit malignant behavior at the cellular level when the effect of TS-PDT is inadequate [26]; the early decline in the PFS curve may be due to inadequate TS-PDT effect, and a certain number of patients with long-term PFS may be due to adequate antitumor effect of TS-PDT. Although the factors that make TS-PDT effective are unclear, the difference in TS-PDT efficacy despite most patients achieving GTR may be related to its effect on tumors lodged in the FLAIR high region or secondary effects such as antitumor immunity [37].

The results of univariate and multivariate analyses suggested that the addition of TS-PDT might be a prognostic factor along with the preoperative KPS, which is a known prognostic factor in recurrent GBM. However, IDH mutation and the pre-recurrence pathology were not prognostic factors. In particular, IDH mutation is known to be a significant prognostic factor in newly diagnosed GBM [38], but its clinical importance in recurrent GBM remains controversial. In the literature on recurrent malignant glioma, there are reports that IDH mutation was associated with a better prognosis [39], whereas there are also reports of better results with IDH wildtype malignant glioma with respect to combined stereotactic radiotherapy with bevacizumab [4]. Contrarily, several publications have reported that IDH mutation is not a prognostic factor at the time of recurrent GBM [40,41], and our results are in agreement with these reports. In view of previous reports and our study results, IDH mutation is a strong prognostic marker of glioma characteristics during the early stage, but it may lose its pivotal role as a prognostic factor when the tumor recurs and pathologically develops as GBM. Taken together, the results of the time-to-event analysis and multivariate analysis suggest that there is benefit in adding TS-PDT to tumor resection when the recurrent tumor is suspected to be a GBM, regardless of its origin or molecular biology.

When photosensitizers accumulate in tumors, peritumoral vessels, or peritumoral tissue stroma, there is debate about which localization determines the impact of PDT’s antitumor effects. Although this has not been fully elucidated in TS-PDT in the brain tumor environment, reports using TS-PDT for tumors in other cancers or using other photosensitizers suggest that accumulation of TS in the tumor itself strongly influences the antitumor effects of TS-PDT, as much as accumulation in peritumor blood vessels [42,43,44]. Our results showed that intra-tumoral accumulation of TS was independent of the pre-recurrence pathology. This result may support that the clinical efficacy of additional PDT for recurrent GBM observed in this study was independent of the patient’s previous pathology.

Concerning the effect of PDT as shown in this study, innovations in drug delivery such as the accumulation of photosensitizers in tumors and peritumoral blood vessels are expected to further enhance this anti-tumor effect, and novel research is being conducted regarding various methods. For example, there are reports on the enhancement of tumor accumulation and antitumor effect by conjugating photosensitizers with various nanoparticles [45], conjugating with tumor-specific antibodies [46], and PDT using a multi-targeted liposome system in which photosensitizers are encapsulated in liposomes targeting the tumor, vascular endothelium, and tumor stroma [47]. With regard to TS, efforts to improve tumor accumulation by incorporating TS into inactivated viral envelopes have recently been reported [48]. The antitumor effect of TS-PDT for recurrent GBM demonstrated in this study may be further improved by devising such a drug-delivery system, and it is expected to evolve as a therapeutic method in the future.

The primary limitation of this study was its retrospective nature. The results of this study were subject to selection bias since all patients underwent surgery for a recurrent lesion. Only recurrent lesions considered removable were eligible for surgery at our institution. Therefore, cases in which surgical removal was difficult due to eloquent lesions or multiple lesions, were excluded.

Despite this limitation, we believe that this study, which demonstrated the efficacy of TS-PDT in treating recurrent GBM, is very valuable with potential clinical application. Future prospective clinical trials are required to confirm the results of this study.

## 5. Conclusions

TS-PDT has significant prognostic value as an adjunct to surgery in treating recurrent GBM. This effect is independent of pre-recurrence pathology and is a versatile additional treatment.

## Figures and Tables

**Figure 1 pharmaceutics-14-00353-f001:**
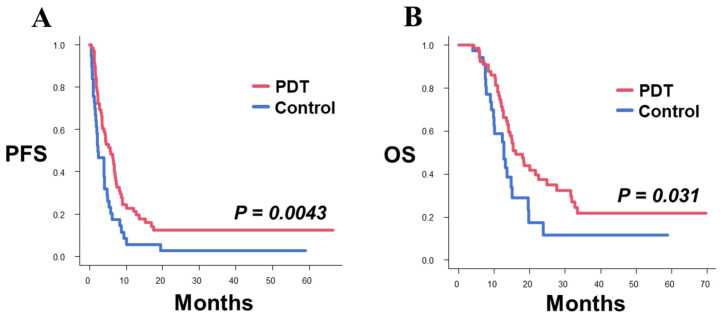
Kaplan–Meier survival curves for PFS and OS after surgery for recurrence in the PDT and control groups. (**A**) Patients in the PDT group showed significantly longer PFS than the patients in the control group (median PFS: PDT 5.7 months, control 2.2 months; *p* = 0.0043). (**B**) Patients in the PDT group showed significantly longer OS than the patients in the control group (median OS: PDT 16.0 months, Control 12.8 months; *p* = 0.031).

**Figure 2 pharmaceutics-14-00353-f002:**
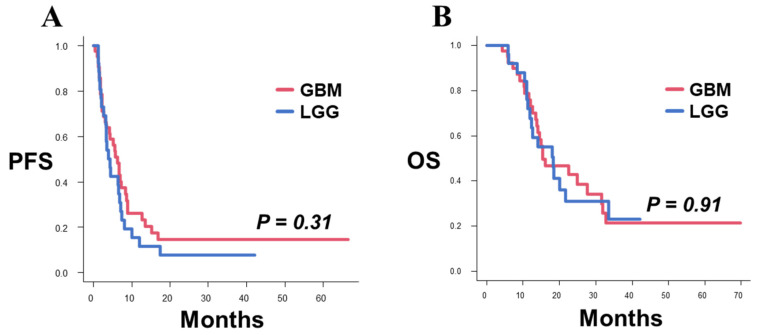
Kaplan–Meier survival curves for PFS and OS after surgery for recurrence in the GBM and LGG groups. (**A**) The PFS was not significantly different between the GBM and LGG groups (median PFS: GBM 6.3 months, LGG 4.2 months; *p* = 0.31). (**B**) The OS was not significantly different between the GBM and LGG groups (median OS: GBM 15.4 months, LGG 18.3 months; *p* = 0.91).

**Figure 3 pharmaceutics-14-00353-f003:**
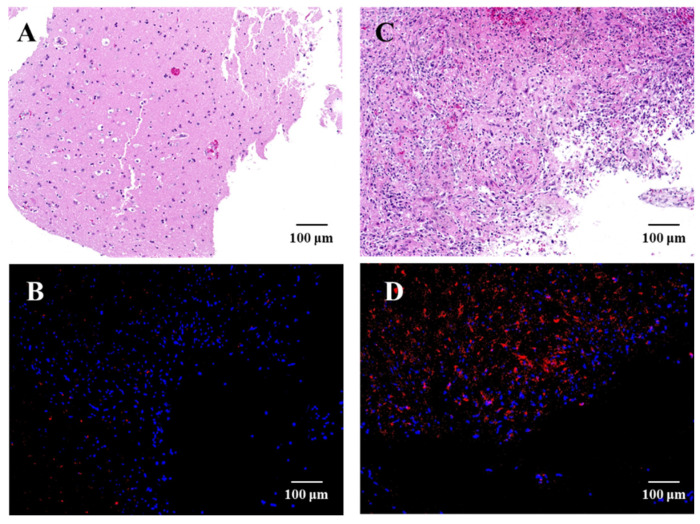
HE staining and fluorescence microscopy images of normal brain tissue surrounding the tumor (**A**,**B**) and recurrent glioblastoma in contrast-enhanced lesions (**C**,**D**). Samples were processed for H&E staining or immunofluorescence examination. The nuclei of the tumor cells were stained with DAPI, and red fluorescence at 640 nm was detected in the tumor cells at an excitation wavelength of 400 nm, which indicated the TS uptake in the tumor and normal brain tissue. Abbreviations: CE, contrast enhanced.

**Figure 4 pharmaceutics-14-00353-f004:**
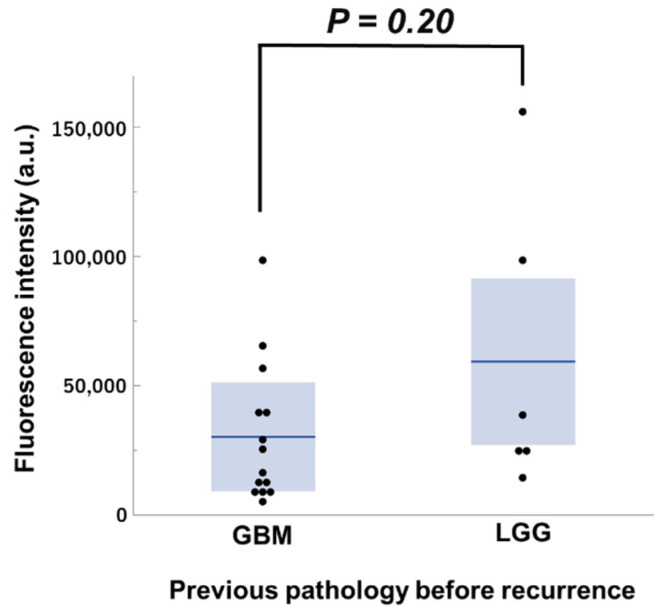
Fluorescence intensity in CE region of recurrent glioblastoma samples from GBM (*n* = 14) and LGG (*n* = 6) groups. There was no significant difference between the two groups (Mann–Whitney U test, *p* = 0.20). Each box represents the interquartile range, and the median was indicated by a bold line. The ends of the whiskers represented the 10th and 90th percentile.

**Figure 5 pharmaceutics-14-00353-f005:**
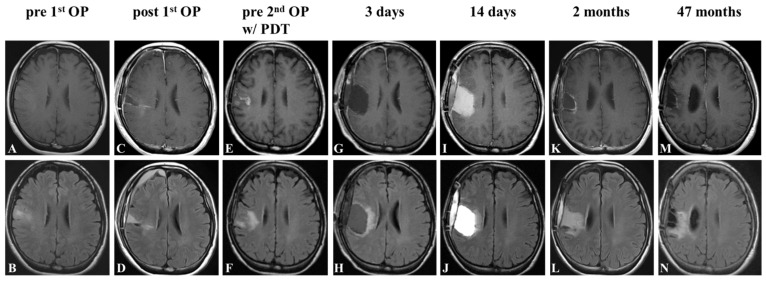
Representative Case 1. Contrast-enhanced T1-weighted and FLAIR MR image before (**A**,**B**) and after (**C**,**D**) the first operation. Contrast-enhanced T1-weighted and FLAIR findings before the second operation with PDT (**E**,**F**); 3 days (**G**,**H**); 2 weeks (**I**,**J**); 2 months (**K**,**L**), and 47 months (**M**,**N**) after the second surgery.

**Figure 6 pharmaceutics-14-00353-f006:**
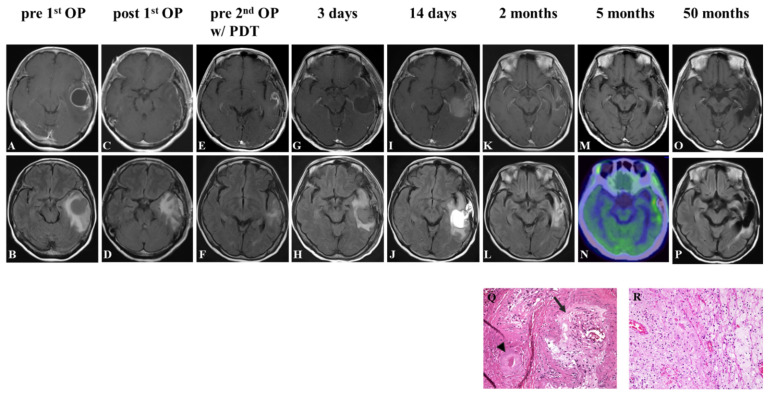
Representative Case 2. Contrast-enhanced T1-weighted and FLAIR MR image before (**A**,**B**) and after (**C**,**D**) the first operation. Contrast-enhanced T1-weighted and FLAIR findings before the second operation with PDT (**E**,**F**); 3 days (**G**,**H**); 2 weeks (**I**,**J**); 2 months (**K**,**L**); 5 months (**M**), and 50 months (**O**,**P**) after the second surgery. PET study at 5 months postoperatively (**N**). The pathological results of the suspected recurrent lesion at 5 months postoperatively showed venous infarction ((**Q**) arrowhead) and loss of internal elastic membrane ((**Q**) arrow), but no obvious tumor cells (**R**).

**Table 1 pharmaceutics-14-00353-t001:** Patient demographics and characteristics.

	PDT	Control	*p* Value
No. of patients	70	38	
Age			
Average (year) ± SD	46.7 ± 13.3	43.4 ± 13.3	0.32
Median (range)	43.5 (20–80)	42 (16–71)	
Sex			0.22
Male	39 (56%)	26 (68%)	
Female	31 (44%)	12 (32%)	
Median preoperative KPS			
KPS score (range)	80 (40–90)	85 (50–100)	0.063
EOR			0.37
GTR	64 (91.4%)	36 (94.7%)	
STR	4 (5.7%)	0	
PR	2 (2.9%)	2 (5.3%)	
Histopathology of rec.			1
GBM	69 (98.6%)	38 (100%)	
Gliosarcoma	1 (1.4%)	0	
Previous pathology before rec.			
GBM	43 (62.3%)	16 (42.1%)	0.071
LGG	26 (37.7%)	22 (57.9%)	
IDH mutation			
Rate of IDH mutation	22.9% (16/70)	39.5% (15/38)	0.11
Mib-1			
Average ± SD	19.8 ± 12.0	23.4 ± 11.4	0.11
Median (range)	17.0 (1.6–51.4)	20.7 (4.0–46.8)	
MGMT protein expression			0.73
High	15 (23.1%)	9 (24.3%)	
Low	28 (43.1%)	13 (35.1%)	
None	22 (33.8%)	15 (40.5%)	

**Table 2 pharmaceutics-14-00353-t002:** Univariate and multivariate analyses for PFS.

Variables		Univariate Analysis		Multivariate Analysis	
		Hazard Ratio [95% CI]	*p* Value	Hazard Ratio [95% CI]	*p* Value
Age	<55y vs. ≥55y	1.52 [0.95–2.44]	0.079		
KPS	<70 vs. ≥70	1.70 [1.04–2.77]	0.035	1.82 [1.11–2.99]	0.017
Pre-rec pathol	GBM vs. LGG	0.76 [0.51–1.15]	0.2		
IDH	mIDH1 vs. wtIDH1	1.42 [0.91–2.23]	0.13		
PDT	PDT + Surgery vs. Surgery alone	0.54 [0.35–0.83]	0.005	0.52 [0.34–0.79]	0.026

Abbreviations: Pre-rec pathol, pre-recurrence pathology; mIDH1, IDH1 mutant type; wtIDH1, IDH1 wild type.

**Table 3 pharmaceutics-14-00353-t003:** Univariate and multivariate analyses for OS.

Variables		Univariate Analysis		Multivariate Analysis	
		Hazard Ratio [95% CI]	*p* Value	Hazard Ratio [95% CI]	*p* Value
Age	<55y vs. ≥55y	1.36 [0.80–2.33]	0.26		
KPS	<70 vs. ≥70	1.79 [1.03–3.09]	0.038	1.82 [1.05–3.15]	0.033
Pre-rec pathol	GBM vs. LGG	1.04 [0.63–1.69]	0.89		
IDH	mIDH1 vs. wtIDH1	1.69 [0.98–2.91]	0.06		
PDT	PDT + Surgery vs. Surgery alone	0.57 [0.34–0.96]	0.034	0.56 [0.33–0.94]	0.029

## Data Availability

The data presented in this study are available on request from the corresponding author.

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
