# Peer review of "Therapeutic Options for Recurrent Glioblastoma—Efficacy of Talaporfin Sodium Mediated Photodynamic Therapy"

_pharmaceutics, 2022, doi:10.3390/pharmaceutics14020353_

Round 1
Reviewer 1 Report
In the manuscript entitled “Therapeutic Options for Recurrent Glioblastoma - Efficacy of Talaporfin Sodium mediated Photodynamic Therapy”, Tatsuya et al. retrospectively evaluated the efficacy of talaporfin sodium (TS) mediated photodynamic therapy (PDT) as a treatment for recurrent GBM. The manuscript compares the prognostic effects of different treatments for patients with recurrent GBM. The results suggest that additional TS-PDT treatment for recurrent GBM could have potential survival benefits. The comments are given as follows.
- In the chapter 3.2, several complications occurred in the PDT group but none in the control group. Please explain this and give the reasons for the reduction of the complications.
- ROS produced by PDT could destroy various cellular organelles and tumor blood vessels, leading to the destruction of tumors. However, this conclusion has not been confirmed in GBM. The author should provide more direct evidence that TS-mediated PDT can also play a similar role in GBM.
- The absorption rate of TS has a great influence on the effect of treatment. However, only the GBM and LGG groups are compared in the manuscript. The author should conduct a more detailed analysis of the influencing factors of TS absorption rate and the relationship between absorption rate and PFS and OS.
Author Response
Thank you for your review and comments.
Please see the attachment.

Reviewer 2 Report
The methodology used to analyze TS accumulation in GBM and normal tissue is referenced but I recommend to add a brief description (additional to the already mentioned).
Also, in Figure 3, please check if is there a mistake in the nomenclature of images. I observe greater red fluorescence in normal tissue than in GBM tissue, and authors claim the opposite in the text (page 7 of 13, line 219-220).
Author Response

(The authors gave the same response as above.)
